# Foveal Hypoplasia in *CRB1*-Related Retinopathies

**DOI:** 10.3390/ijms241813932

**Published:** 2023-09-11

**Authors:** Ana Catalina Rodriguez-Martinez, Bethany Elora Higgins, Vijay Tailor-Hamblin, Samantha Malka, Riccardo Cheloni, Alexander Mark Collins, John Bladen, Robert Henderson, Mariya Moosajee

**Affiliations:** 1UCL Institute of Ophthalmology, London EC1V 9EL, UK; ana.rodriguezmartinez@nhs.net (A.C.R.-M.); bethany.higgins@ucl.ac.uk (B.E.H.); vijaytailor1@nhs.net (V.T.-H.); samantha.malka@nhs.net (S.M.); r.cheloni@ucl.ac.uk (R.C.); amc122@ic.ac.uk (A.M.C.); robert.henderson@gosh.nhs.uk (R.H.); 2Moorfields Eye Hospital NHS Foundation Trust, London EC1V 2PD, UK; 3Great Ormond Street Hospital for Children NHS Foundation Trust, London WC1N 1LE, UK; 4UCL Experimental Psychology, London WC1H 0AP, UK; 5King’s College Hospital NHS Foundation Trust, Strand, London WC2R 2LS, UK; john.bladen1@nhs.net; 6The Francis Crick Institute, London NW1 1AT, UK

**Keywords:** retinal dystrophy, LCA, early-onset severe retinal dystrophy, retinitis pigmentosa (RP), macular dystrophy (MD), cone-rod dystrophy (CORD), foveal hypoplasia (FH), optical coherence tomography (OCT), Crumbs cell polarity complex component 1 gene (*CRB1*)

## Abstract

The *CRB1* gene plays a role in retinal development and its maintenance. When disrupted, it gives a range of phenotypes such as early-onset severe retinal dystrophy/Leber congenital amaurosis (EOSRD/LCA), retinitis pigmentosa (RP), cone-rod dystrophy (CORD) and macular dystrophy (MD). Studies in *CRB1* retinopathies have shown thickening and coarse lamination of retinal layers resembling an immature retina. Its role in foveal development has not yet been described; however, this retrospective study is the first to report foveal hypoplasia (FH) presence in a *CRB1*-related retinopathy cohort. Patients with pathogenic biallelic *CRB1* variants from Moorfields Eye Hospital, London, UK, were collected. Demographic, clinical data and SD-OCT analyses with FH structural grading were performed. A total of 15 (48%) patients had EOSRD/LCA, 11 (35%) MD, 3 (9%) CORD and 2 (6%) RP. FH was observed in 20 (65%; CI: 0.47–0.79) patients, all of whom were grade 1. A significant difference in BCVA between patients with FH and without was found (*p* = 0.014). BCVA continued to worsen over time in both groups (*p* < 0.001), irrespective of FH. This study reports FH in a *CRB1* cohort, supporting the role of *CRB1* in foveal development. FH was associated with poorer BCVA and abnormal retinal morphology. Nonetheless, its presence did not alter the disease progression.

## 1. Introduction

Biallelic pathogenic variants in the Crumbs cell polarity complex component 1 gene (*CRB1*, OMIM #604210) result in a diverse spectrum of retinopathies with phenotypic variability. The most common phenotype reported is Leber congenital amaurosis (OMIM #613935, LCA8), accounting for 7–17%, followed by 3–9% of autosomal recessive retinitis pigmentosa (OMIM #600105, RP12), then cone-rod dystrophy (CORD) and macular dystrophy (MD) [1,2,3]. Distinctive features of *CRB1* retinopathies are nummular pigmentation, fine yellow punctate deposits, preserved para-arteriolar retinal pigment epithelium (PPRPE) and coarse and thickened retina [4], with some cases presenting with a Coats-like vasculopathy [5]. More than 300 causative variants have been reported to cause *CRB1*-related retinopathies. The only genotype–phenotype correlation known to date is the in-frame deletion c.498_506del p.Ile167_Gly169del associated with MD [6,7], which presents with a pattern of degeneration affecting the superior, inferior and nasal retina to the optic nerve head, similar to that seen in *ADAM9* and *CDH3* maculopathy [7].

The *CRB1* gene encodes a type 1 transmembrane protein consisting of 19 epidermal growth factor (EGF) domains, 3 laminin-globular (LamG) like extracellular domains and a short FERM/PDZ blinding motif containing intracellular cytoplasmic tail, which is localised to the subapical region of Müller glia and photoreceptor cells. *CRB1* has a role in retinal development and long-term retinal integrity. Its main function is to maintain the zonula adherence junctions at the external limiting membrane (ELM) and thus contributes to vascular integrity and apicobasal polarity. Although the underlying disease mechanism of CRB1 retinopathy remains unclear, a recent study using the oko meduzy (crb2a^m289^) zebrafish model and human retinal organoids derived from LCA8 patient-specific induced pluripotent stem cells (hiPSCs) showed abnormal regulation in transcriptional and cell cycle pathways, and epigenetic pathways with global correlation between the transcriptome and methylation status. An increase in VSX2 and PAX6 expression in LCA8 patient retinal organoids suggested an enriched retinal progenitor cell (RPC) population with significantly higher cell proliferation with inhibited cell fate progression, confirmed by a reduction in BRN3A-positive retinal ganglion cells (RGCs) [6]. The RPCs lose polarity and adherence, resulting in cell detachment and causing severe retinal laminae disorganisation, which can be detected by optical coherence tomography (OCT) imaging [1,2,3]. Despite the evidence of the role of *CRB1* in retinal development, its role in foveal development has not yet been described.

Normal human foveal development begins at 22 weeks gestation and continues until around 45 months of age, according to histologic and OCT imaging studies [8,9,10]. It is characterised by centrifugal displacement of inner retinal layers, cone photoreceptor specialisation and centripetal migration of cone photoreceptors, and genes such as *PAX6*, *SLC38A8* and *TYR* are known to be involved [11,12]. OCT enables morphologic visualisation of these stages, as it shows the formation and deepening of the foveal pit with outward displacement of the plexiform layers, outer segment lengthening and outer nuclear layer widening. Failure of any of these processes results in foveal hypoplasia (FH) [8,13]. Structural grading of FH using OCT imaging is widely used and includes grades 1 to 4, representing the most to least developed fovea, whereas the atypical grade represents a disruption of the outer retinal layers [9,13].

FH has been described in various ocular conditions, including albinism, mutations in genes such as *SCLC38A8*, *PAX6*, *FRMD7* and *AHR* and achromatopsia [12,14,15,16,17]. Additionally, it has been reported in cases of choroideremia (CHM) with no apparent connection to developmental pathways and in unaffected central visual acuity [18,19]. FH has not yet been reported in *CRB1* retinopathies. This study reports the prevalence and grades of FH seen in a molecularly confirmed *CRB1* cohort with biallelic pathogenic variants, supporting the role of *CRB1* in normal foveal development.

## 2. Results

### 2.1. Demographics

Sixty-nine patients with molecularly confirmed *CRB1*-related retinopathy from 63 unrelated families were identified from the Moorfields Eye Hospital, London. Forty-four patients had good-quality OCT scans available. A total of 13 patients were excluded from the FH analysis: 6 due to cystoid macular oedema (CMO), 7 due to severe macular atrophy (pseudocoloboma) and 1 due to epiretinal membrane (ERM), leaving a total of 31 patients for the analysis. Ethnicity information was available in 16 patients, with 14 identified as white, 2 as black and 1 as Asian. Details of the demographic characteristics of this cohort are reported in Table 1.

### 2.2. Clinical and FH Classification

Based on clinical data, retinal imaging and age of onset, 15 (48%) patients were diagnosed as EOSRD/LCA, 11 (35%) as MD, 3 (9%) as CORD and 2 (6%) as RP. The mean age of onset was 4.2 ± 1.9 years old for EOSRD/LCA, 15 ± 13.7 years for MD, 11.9 ± 12.4 years for CORD and 13.6 ± 13.7 years for RP.

FH structural grading was performed on 62 eyes of 31 patients and was observed in 20 (65%; CI: 0.47–0.79) patients, of whom 12 (60%) had EOSRD/LCA, 5 (25%) MD, 2 (10%) RP and 1 (5%) CORD. Of those with FH, 15 (75%) had grade 1a, and 5 (25%) had grade 1b. A total of 11 (35%; CI: 0.21, 0.53) patients did not have FH, of whom 3 had EOSRD/LCA (27%), 6 (56%) had MD and 2 (18%) had CORD. Using the FH structural grading scheme, the intergrader kappa correlation coefficient was 0.91 when identifying FH, suggesting almost perfect agreement, and 0.74 when classifying grade 1 into either 1a or 1b, suggesting that the agreement is substantial. Each case of disagreement was discussed over what constituted a nearly normal pit as per grade 1a versus a shallow pit as per grade 1b; total agreement ensued after discussion and training. Examples of multimodal imaging in representative patients with and without FH in *CRB1* retinopathies are reported in Figure 1 and the Appendix A. The mean age of the cohort was 31 years (SD ± 16 years), 34 years (SD ± 17 years) in those with FH, and 25 years (SD ± 14 years) in those without. No significant difference was observed between groups (*p* = 0.244).

### 2.3. Visual Acuity and Refraction

Mean BCVA was 1.13 logMAR (SD ± 0.88 logMAR) for the entire cohort, 1.42 logMAR (SD ± 0.89 logMAR) for patients with FH and 0.63 logMAR (SD ± 0.62 logMAR) in those without. A significant difference between patients with and without FH BCVA was found (*p* = 0.014), and there was greater variability in BCVA scores in patients with FH (Figure 2A). The mean spherical equivalent (SE) in the FH group was +2.30 (SD ± 3.8) on the right eye and +2.25 (SD ± 3.9) on the left eye. While in those without FH, it was +1.47 (SD ± 2.90) on the right eye and +1.11 (SD ± 2.5) on the left eye.

A total of 15 patients underwent more than one visit where BCVA was recorded, ranging from 2 months to 46 years from the baseline visit. Among them, there were 8 with FH (6 EOSRD/LCA and 2 MD) and 7 without FH (2 EOSRD/LCA and 5 MD). The mean BCVA decline from baseline to the most recent measure was 0.05 logMAR for the whole cohort, 0.06 (SD ± 0.51) logMAR for patients with FH and 0.042 (SD ± 1.2) logMAR for those without. A linear mixed model of the data revealed a significant association between BCVA and time, suggesting that BCVA worsened as time progressed (*p* < 0.001). However, FH did not show a significant association with BCVA worsening in this model (see Figure 2B).

### 2.4. Quantitative and Qualitative OCT Imaging

Quantitative OCT assessment was possible in 26 patients, including 18 with FH (11 EOSRD/LCA, 4 MD, 2 RP and 1 CORD) and 8 without FH (2 EOSRD/LCA, 4 MD and 2 CORD). There was an increase in thickness and volume of the fovea, inner ring thickness (IRT) and inner ring volume (IRV) in patients with FH, with no statistically significant differences between the two groups. There was a statistically significant difference between the two groups in the temporal inner volume (*p* = 0.048) (Table 2). Normative data from the general population and details of OCT thickness and volume in our cohort are reported in Figure 3A.

Qualitative OCT assessment was performed in 31 patients. Among the 20 patients with FH, 9 (45%) had coarse lamination (group 2), 11 (55%) had disorganisation with coarse lamination (group 3) and none had normal retinal lamination (group 1). Conversely, of the 11 patients without FH, 6 (54%) had normal retinal lamination (group 1), 4 (36%) had coarse lamination (group 2) and only 1 (9%) had disorganisation with coarse lamination (group 3) (Figure 3B). A comparison between normal retinal lamination (group 1) and abnormal lamination (groups 2 and 3) was performed, showing no statistically significant association with FH (*p* = 0.683).

More than one OCT scan was available for 22 patients, and these ranged over a period of 3 months to 13 years from the baseline visit: 16 patients in the FH group (11 EOSRD/LCA, 3 MD, 1 RP and 1 CORD), and 6 in the group without FH (1 EOSRD/LCA, 4 MD and 1 CORD). See Table 3 for the mean change in OCT parameters over time. Following mixed linear modelling, the rate of change per year was not significantly different between those with FH and those without for foveal thickness/volume or mean inner ring thickness/volume.

### 2.5. Molecular Characteristics

Genetic analyses of this cohort are shown in Table 1. In total, 34 different variants were identified, of which 21 were missense, 1 in-frame deletion, 4 frameshift, 4 splice and 1 exon deletion; all were likely pathogenic or pathogenic. The missense variant c.2843G>A p.(Cys948Tyr) and the in-frame deletion c.498_506del p.(Ile167_Gly169del) were the most prevalent disease-causing *CRB1* variant in our cohort (*n* = 7). Six out of the seven patients who had the c.2843G>A variant presented with the EOSRD phenotype. All patients who presented the in-frame deletion c.498_506del p.(Ile167_Gly169del) had the macular dystrophy phenotype. No specific variant was associated with the presence of foveal hypoplasia.

## 3. Discussion

This retrospective study is the first to report the presence of FH in a molecularly confirmed *CRB1*-related retinopathy cohort following qualitative and quantitative SD-OCT analyses. FH was seen in 65% of patients, with a higher prevalence in, but not limited to, the EOSRD/LCA phenotype. The *CRB1* gene plays an important role in both retinal development and the long-term maintenance of retinal integrity [20], being required for apical–basal cell polarity and adhesion between photoreceptors and Müller glial cells [6,20,21,22]. In vitro models of *CRB1* retinopathies in mouse (*Crb1*^−/−^) have shown a loss of integrity at the subapical region–adherence junctions at the outer limiting membrane with displaced photoreceptors in the subretinal space [23]. Similarly, retinal changes seen within zebrafish *crb2a*^−/−^ led to the absence of retinal layer demarcation, complete cellular disorganisation with patches of plexiform matter, cell proliferation, reduced cell cycle exit and lack of neuronal differentiation, all of which resembles an immature retina [6]. Despite the evidence of the role of *CRB1* in retinal development, its role in foveal development has not yet been described. Normal foveal development involves three main events, including centrifugal displacement of the inner retinal layers, cone photoreceptor specialisation and centripetal migration of cone photoreceptors and failure of any of these processes results in foveal hypoplasia [8,9]. It is possible that increased cell numbers and poor/delayed cone specialisation could be contributing to FH in *CRB1*-retinopathy, but further disease modelling and earlier deep phenotyping of patients would be required to confirm this.

A multicentre study of 907 patients with FH found that 67% of individuals had albinism caused by variants in *GPR143*, *OCA2*, *TYR* and *HPS1* genes, followed by 21.8% with *PAX6*, 6.8% with *SLC38A8* and 3.5% with *FRMD7* variants [16]. Each gene exerts a different mechanism to disrupt foveal development. The *PAX6* gene, considered the master regulator of the eye, is expressed in both neural retina and RPE by week five of human gestation and in retinal ganglion, amacrine and horizontal cells, lens, cornea, conjunctiva, iris and ciliary body postnatally. It has two DNA-binding domains, the paired domain (PD) and the homeodomain (HD), connected by a linker region. Both subdomains bind the respective consensus DNA sequences, and the two major *PAX6* isoforms, canonical *PAX6* and PAX6(5a), modulate their activity [12,24]. The PAX6(5a) isoform is highly expressed in the fovea, and mutations in exon 5a, which affect the C-terminal subdomain (CTS) binding activity, are known to cause FH [24]. *PAX6* mutations are associated with a wider spectrum of FH, ranging from grades 1–4 [12]. Genes associated with albinism are involved in melanogenesis, converting tyrosine to L-DOPA (a precursor of melanin) via the enzyme tyrosinase. The process is crucial for the production of retinal pigmentation, the metabolism of retinal ganglion cells and the organisation and projection of retinal–fugal fibres; abnormalities of these processes can cause FH [15]. Albinism has been associated with FH grades 2 or worse FH in most patients [9]. The *SLC38A8* gene encodes an orphan member of the SLC38 sodium-coupled neutral amino acid transporter (SNAT) family of proteins, which are widely expressed and predominantly have glutamine as their preferred substrate. The SLC38A8 glutamine transporter is found throughout the neural retina with particularly strong expression in the inner and outer plexiform layer and photoreceptor layer. The presence of SLC38A8 in the retina suggests its involvement in synaptic neurotransmitter recycling, which contributes to the formation and remodelling of neural circuits, including the projecting retinal ganglion cells [11,14]. Patients with *SLC38A8* have more extensive FH with grades 3–4, suggesting an earlier arrest in foveal development [9,16]. The FRMD7 protein, a member of the FERM family of proteins associated with cytoskeletal dynamics, is involved in the elongation of neurites during neuronal development [25]. Mutations in the *FRMD7* gene influence the maturation and complexities of neuronal processes, potentially involving Rho GTPase signalling [25] and are associated with grade 1 FH, indicating that foveal development is affected later in the modelling process [16]. Among the observed FH cases in our *CRB1* cohort, all were classified as grade 1. This suggests a potentially delayed or late effect on fovea development.

A direct correlation between BCVA and the extent of FH has been described, with worse acuity observed in higher or atypical FH gradings [9,13,16]. In this study, a significant difference in BCVA between patients with FH compared with those without (*p* = 0.014) was observed, despite all FH patients having mild grading scores (grade 1). It is noteworthy to highlight that the FH group exhibited a higher prevalence of the EOSRD/LCA phenotype, which is associated with poorer visual acuity [3]. Although FH may be contributing to the baseline vision, there will be other factors having a more significant impact such as the maturation state of the retina and active degenerative processes underway. Due to the small sample size, it was not possible to compare the presence versus absence of FH within the EORSD/LCA phenotype group. The mean BCVA of our cohort was 1.13 LogMAR (SD ± 0.88 LogMAR), poorer compared with the reported BCVA values of 0.88 LogMAR and 0.7 LogMAR in studies in which most of their cohort included the RP phenotype [1,2]. However, our BCVA results were better when compared with a similar cohort, which included mostly EOSRD/LCA patients, as they reported a BCVA of 1.6 logMAR (SD ± 0.88 LogMAR) [3]. The observed difference could be attributed to the exclusion of patients of this cohort with poor imaging (e.g., those with nystagmus and corneal or lens opacities), CMO, ERM and severe macular atrophy, which are associated with poorer visual acuity. Published cohort studies have reported significant differences from baseline and at the 9-year follow-up, with rates of BCVA decline of 0.06 LogMAR in EOSRD/LCA, 0.07 LogMAR in RP and 0.04 LogMAR in MD per year [3] and reported that 70% of patients reach blindness, defined as visual acuity worse than 3/60 based on the World Health Organization criteria [2,3]. Our cohort exhibited a mean BCVA decline over time of 0.05 LogMAR, with similar progression rates seen in both FH and those without, indicating that FH does not seem to alter disease progression.

Retinal thickening and abnormal retinal architecture have been some of the most consistent findings in *CRB1*-associated retinopathies [1,2,3,4,7]. In a *CRB1* cohort study, 76% of patients had abnormal retina, including coarse lamination with or without disorganisation [1]. Another cohort found that *CRB1* EOSRD/LCA had a higher prevalence of abnormal retinal morphology when compared with the other phenotypes [3]. All patients from our cohort with FH had abnormal retinal architecture (groups 2 and 3), whereas 54% of patients without FH displayed normal retina on OCT (group 1). Additionally, FH patients demonstrated an increased thickness and foveal volume, inner ring thickness (IRT) and inner ring volume (IRV). Despite differences in thickness and volume between the FH group and that without, the outer retinal bands, including the ellipsoid zone and external limiting membrane, were either attenuated or unidentifiable in the fovea and perifovea in both groups, owing to the rate of disease progression at an early age in *CRB1*-associated retinopathies, which impeded quantitative analysis [1]. Furthermore, pseudocoloboma was noticed in 15% of the original sample, mainly in the LCA/EOSRD phenotype with a severe presentation. This clinical presentation has been previously described [2] and associated with LCA due to mutations in *CRX*, *AIPL* and *NMNAT1* [26,27]. A noteworthy observation is the possibility of abnormal foveal development in the absence of FH, suggesting the presence of subtle developmental abnormalities that could make the fovea susceptible to degeneration, as depicted in Figure 1B. This phenomenon of foveal maldevelopment is seen in *NMNAT1*-associated cone-rod dystrophy [26].

The most common variants in our *CRB1* cohort were the missense c.2843G>A, p.Cys948Tyr and the in-frame deletion c.498_506del, p.Ile167_Gly169del. Six out of the seven EOSRD/LCA patients had the c.2843G>A, p.Cys948Tyr variant; this has also been denoted as most frequently occurring in other *CRB1*-EOSRD cohorts in European populations [3,28]. Comparably, seven patients who had the in-frame deletion c.498_506del, p.Ile167_Gly169del had the MD phenotype, which has been previously associated with this phenotype [2,7]. No particular variant was linked to FH.

Potential treatments are under development. Preclinical studies using patient-derived retinal organoids derived from *CRB1* RP patients with different variants, including (i) c.3122T>C p.(Met1041Thr) homozygous missense mutation, (ii) compound heterozygous c.2983G>T p.(Glu995*) and c.1892A>G, p.(Tyr631Cys) and (iii) compound heterozygous c.2843G>A p.(Cys948Tyr) and c.3122T>C p.(Met1041Thr) variants, have demonstrated restoration of the histological retinal phenotype, showing an increased number of photoreceptor nuclei and fewer photoreceptor nuclei protruding above the outer limiting membrane (OLM) in the treated group using AAV-mediated gene augmentation compared with the control group [29]. In alternative approaches, interventions were undertaken on Crb1Crb2^F/+^ cKO and Crb2 cKO *CRB1*-RP mouse models, which showed impairment of retinal function and structure postnatally from 1 and 3 months onwards, respectively. These interventions, conducted at the midstage of the disease, targeted *CRB2* in both Müller glial cells and photoreceptors and prevented further loss of retinal function [23]. However, the RP phenotype differs from EOSRD/LCA-associated *CRB1*-retinopathy, which displays a more severe retinal dysplasia. The findings of a higher prevalence of FH in the EOSRD/LCA group, together with increased retinal thickness and disrupted lamination, suggest that for more precision medicine, consideration of the varying clinical phenotypes, therapeutic strategy and window of treatment opportunity are essential for optimising the response/outcomes.

### Limitations

*CRB1* retinopathies are rare inherited retinal diseases, and although our study is amongst the largest longitudinal case series, the sample size remains small, with limitations in statistical power and sample heterogeneity. There were no records of axial length on the database to review whether there was a difference between the FH and no FH groups, as nanophthalmos has been reported in *CRB1* retinopathies [30]. Data for this study were retrospectively collected from routine hospital visits, and this may have resulted in selection bias, with analysis of only 31 scans out of the total cohort, variable test types and frequency of examinations between patients. Additionally, *CRB1* retinopathies have coarse and thickened retinas, which make structural OCT grading for FH more difficult. To overcome this, two experienced ophthalmologists underwent a training session and a discussion about the FH structural OCT classification described by Thomas et al. [13]. The high rates of reported FH may be due to the chosen methodology and the association between *CRB1* variants, resulting in thicker retinas [2,3,31]. Quantitative measurement of retinal layers was attempted but could not be performed since 54% of this cohort had abnormal retinal lamination, which impedes accurate categorisation [32].

## 4. Materials and Methods

### 4.1. Subjects

A retrospective observational study at a single tertiary referral centre (Moorfields Eye Hospital NHS Foundation Trust, London, UK). Potential subjects were identified from the prospectively consented Moorfields Eye Hospital Inherited Eye Disease Database for structure/function of genetic diseases (Research Ethics Number: 12/LO/0141), and all procedures adhered to the tenets of the Declaration of Helsinki. Data for these studies are collected as part of standard of care and retrospectively analysed. The inclusion criteria were to have molecularly confirmed biallelic (pathogenic or likely pathogenic) variants in *CRB1*.

### 4.2. Genetics

The methodology of genetic testing and variant interpretation at Moorfields has been described previously [33]. DNA samples extracted from peripheral blood with informed consent were used for genetic testing. Molecular testing was performed in the clinical and research setting, using NGS panel testing through the Rare & Inherited Disease Genomic Laboratory at Great Ormond Street Hospital (GOSH) and whole genome sequencing (WGS) as part of the UK Genomics England 100,000 Genomes Project, where the results were reviewed by a multidisciplinary team to confirm variant pathogenicity, prevalence in publicly available genome databases, the clinical phenotype and mode of inheritance before the molecular diagnosis was established [34].

### 4.3. Clinical

Demographics, clinical data, past medical and ophthalmic history, refractive error, fundoscopy and best-corrected visual acuity (BCVA) were collected from full ophthalmic assessments conducted at each visit as part of their routine clinical care. Patients were categorised into different phenotypes based on clinical data, retinal imaging and age of onset, including early-onset severe retinal dystrophy/Leber congenital amaurosis (EOSRD/LCA), macular dystrophy (MD), cone-rod dystrophy (CORD) and retinitis pigmentosa (RP). BCVA was converted to logarithmic minimum angle of resolution (LogMAR) for statistical analysis. Count fingers vision was given a value of LogMAR 1.98, and hand motion, LogMAR 2.28, light perception and no light perception were LogMAR 2.7 and 3, respectively.

### 4.4. Retinal Imaging

Macular SD-OCT scans were conducted with Spectralis (Heidelberg Engineering, Heidelberg, Germany) in a 6 mm^2^ area that included the standard 1, 3 and 6 mm grid template from the ETDRS. Macular cube scans used for FH structural grading and quantitative analysis were performed with varying scan patterns: either 19 B-scans (512 A-scans/B-scans) or 97 B-scans (1024 A-scans/B-scans) centred on the fovea. Segmentation of SD-OCT images was performed with the integrated automatic segmentation Spectralis software (Heyex Version 2), and errors in the segmentation were manually corrected. Patients without SD-OCT records, low-quality SD-OCT imaging preventing proper grading (e.g., nystagmus and corneal or lens opacities), or with additional macular pathologies such as cystoid macular oedema (CMO), epiretinal membrane (ERM) and severe macular atrophy (pseudocoloboma) were excluded from the OCT analysis. FH structural grading was performed by two independent, experienced ophthalmologists (ARM, JB) following a training session and according to the classification described by Thomas et al. [13]. Grade 1 was defined by the absence of extrusion of plexiform layers, presence of outer nuclear layer (ONL) widening and outer segment (OS) lengthening and either nearly normal foveal pit (grade 1a) or shallow foveal pit (grade 1b). Grade 2 was defined by all features of grade 1 and absence of a foveal pit. Grade 3 was defined by all features of grade 2, plus the absence of OS lengthening. Grade 4 was defined by all features of grade 3, plus the absence of ONL widening at the fovea [13]. Retinal organisation and lamination on SD-OCT were graded as follows: group 1, normal; group 2, normal organisation with coarse lamination; and group 3, disorganisation with coarse lamination. For the quantitative analysis, thickness and volume data were taken from the central portion and the 3 mm inner ring portion of the ETDRS grid. The outer ring was not included in this analysis due to data missingness. Foveal thickness was defined as the average thickness in the central 1000 μm diameter from the internal limiting membrane (ILM) to Bruch’s membrane in a 1 mm diameter circle centred on the fovea of the EDTRS layout. Central foveal thickness was defined as the mean thickness at the point of intersection of the six radial scans [35]. Additionally, we reported inner ring volume (IRV) and inner ring thickness (IRT). Wide-field colour fundus photos were collected with Optos California (Optos plc, Dunfermline, UK).

### 4.5. Statistical Analysis

As both eyes were found to be functionally and structurally equivalent, the right eye was chosen as the study eye for all analyses. The agreement between two independent graders for FH was assessed with the weighted κ-coefficient. The agreement was considered poor if κ < 0.40, moderate if κ = 0.4 to 0.59, substantial if κ = 0.6 to 0.79 and excellent if κ ≥ 0.80. Data were not normally distributed, so nonparametric testing was used for the comparison of means. Bonferroni correction was used to adjust for multiple comparisons. A linear mixed model analysis was used to examine the relationship between OCT parameters and the predictors of FH and time while considering the potential influence of different patients as a random effect. The model was fitted using the restricted maximum likelihood (REML) method, and 95% proportional confidence intervals (CI) were given using the Wald method. Normative data from 19 patients aged 20–40 years for OCT thickness measures were sourced from Grover et al. [36] and compared with this dataset. OCT volume parameters were taken from 50 subjects from Murthy et al. to characterise our data [37]. This approach has been previously conducted in a cohort with *CRB1* by Varela [3]. All analyses were calculated using R, including *irr*, *lme4* and *ggplot2* packages (version 3.3.0; R Foundation for Statistical Computing, 2016).

## 5. Conclusions

In conclusion, this study reports FH in a molecularly confirmed *CRB1* cohort, supporting the role of *CRB1* in foveal development. FH was found to be associated with poorer visual acuity and abnormal retinal morphology. Nonetheless, its presence did not alter the progression of the disease. As we make further progress in therapeutic development and the search for effective biomarkers to monitor response, the significance of understanding the role of genetic variations on clinical presentation in *CRB1* patients will continue to be of paramount importance.

## Figures and Tables

**Figure 1 ijms-24-13932-f001:**
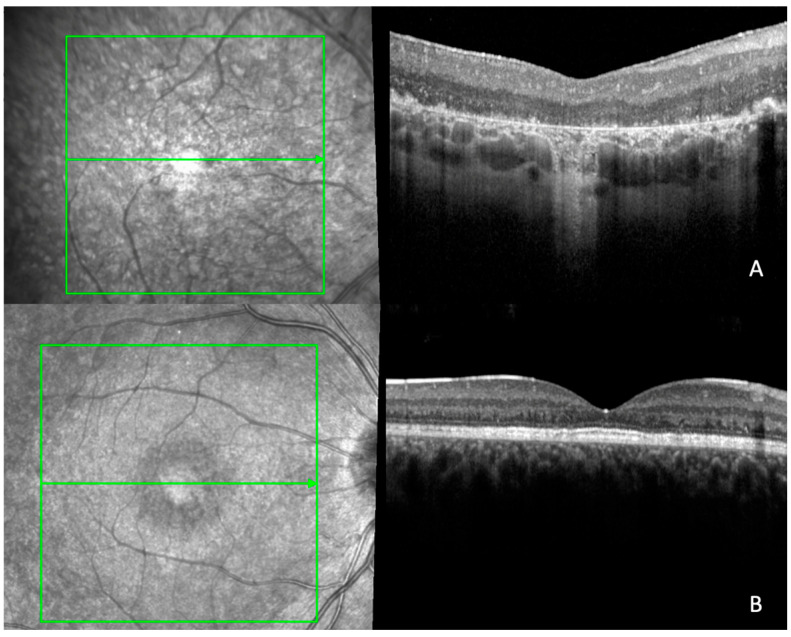
Multimodal imaging in representative patients: (**A**) 19-year-old patient with EOSRD/LCA with grade 1b FH; (**B**) 17-year-old patient with MD without FH.

**Figure 2 ijms-24-13932-f002:**
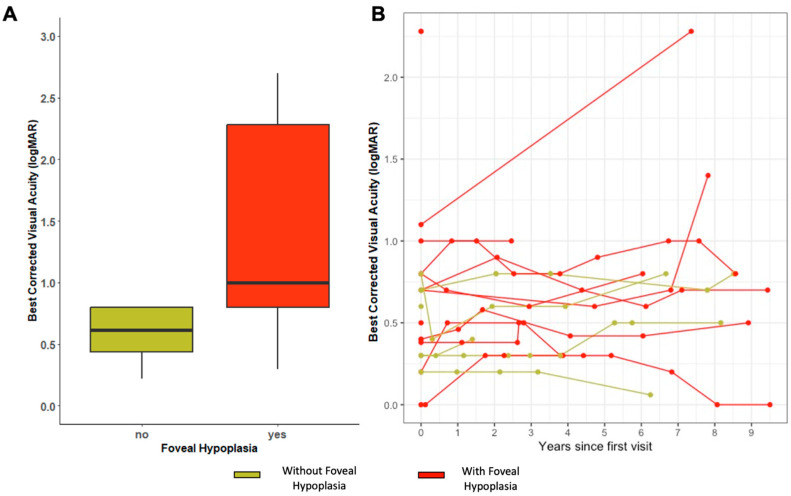
(**A**) Best-corrected visual acuity (BCVA) scores in groups with and without FH. (**B**) Best-corrected visual acuity (BCVA) scores over a 10-year follow-up from baseline visit in both groups.

**Figure 3 ijms-24-13932-f003:**
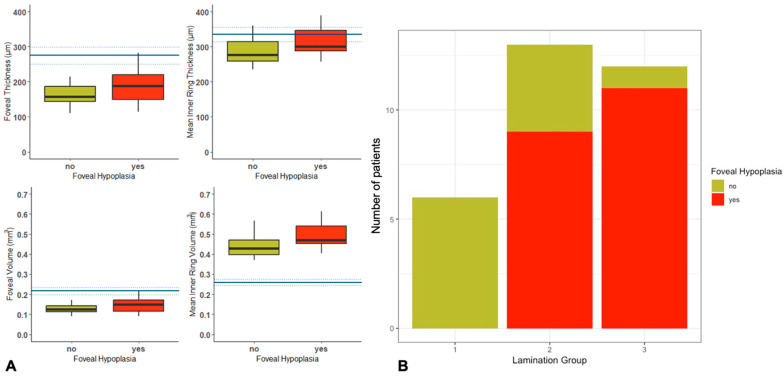
(**A**) Comparison of quantitative OCT imaging with normative data, showing an increase in thickness and volume of the fovea, inner ring thickness (IRT) and inner ring volume (IRV) in patients with FH, with no statistically significant differences between the two groups. (**B**) Retinal lamination showing a higher prevalence of worse retinal lamination (groups 2 and 3) in the FH group than those without FH.

**Table 1 ijms-24-13932-t001:** Summary of subject demographics, genetic results and clinical characteristics of the 31 patients with biallelic pathogenic variants in *CRB1*.

Family Number	Subject	Gender	Ethnicity	Age	Phenotype	Zygosity	Variant 1 cDNA Variant 1 Protein	Variant 1 cDNA Variant 1 Protein	FH
3760	01	M	Asian–Pakistani	37	RP	Homozygous	c.2536G>A p.Gly846Arg		1a
16291	02	M	White	31	EOSRD/LCA	Compound Heterozygous	c.2290C>T p.Arg764Cys	c.2401A>T p.Lys801Ter	1a
20059	03	M	Unknown	53	EOSRD/LCA	Homozygous	c.1831T>C p.Ser611Pro		1b
24835	04	M	White	24	EOSRD/LCA	Compound Heterozygous	c.2129A>T p.Glu710Val	c.3988del p.Glu1330Serfs*11	1b
24839	05	M	Unknown	47	MD	Compound Heterozygous	c.498_506del p.Ile167_Gly169del	c.584G>Tp.Ile167_Gly169del	1a
19331	06	F	Unknown	23	EOSRD/LCA	Compound Heterozygous	c.2843G>A p.Cys948Tyr	c.1712A>C p.Glu571Ala	1a
17595	07	M	Unknown	19	EOSRD/LCA	Compound Heterozygous	c.2043T>A p.Cys681Ter	c.2843G>A p.Cys948Tyr	1b
1826	08	M	White	52	EOSRD/LCA	Compound Heterozygous	c.2129A>T p.Glu710Val	c.2843G>A p.Cys948Tyr	1a
2824	09	M	White	24	EOSRD/LCA	Compound Heterozygous	c.2129A>T p.Glu710Val	c.2234C>T p.Thr745Met	1a
2824	10	F	White	17	EOSRD/LCA	Compound Heterozygous	c.455G>A p.Cys152Tyr	c.3014A>T p.Asp1005Val	1a
20745	11	M	Black	19	EOSRD/LCA	Compound Heterozygous	c.988+1G>T N/A	c.1183G>T p.Glu395Ter	1a
30437	12	M	Unknown	9	EOSRD/LCA	Compound Heterozygous	c.2843G>A p.Cys948Tyr	c.1712A>C p.Glu571Ala	1b
24605	13	M	Unknown	12	MD	Compound Heterozygous	c.2234C>T p.Thr745Met	c.2506C>A p.Pro836Thr	1a
26081	14	F	Black	38	MD	Compound Heterozygous	c.2506C>A p.Pro836Thr	Del exon 6	1a
3642	15	M	Unknown	36	EOSRD/LCA	Homozygous	c.750T>G p.Cys250Trp		1b
4441	16	M	White	40	RP	Homozygous	c.2639A>G p.Asn880Ser		1a
11201	17	M	White	76	MD	Compound Heterozygous	c.253T>C p.Cys85Arg	c.4009_4015del p.Ala1337Thrfs*2	1a
20409	18	M	Unknown	36	MD	Compound Heterozygous	c.584G>T	c.2843G>A p.Cys948Tyr	1a
4441	19	M	White	61	CORD	Homozygous	c.2639A>Gp.N880S		1a
4441	20	M	White	28	MD	Compound Heterozygous	c.498_506del p.Ile167_Gly169del	c.2688T>A p.Cys896Ter	1a
Z413096	21	F	Unknown	11	MD	Homozygous	c.2506C>Ap.Pro836Thr		No
Z889804	22	F	Unknown	13	CORD	Compound Heterozygous	c.498_506del p.Ile167_Gly169del	c.4005 + 1G>A N/A	No
29543	23	F	Unknown	42	MD	Compound Heterozygous	c.2234C>T p.Thr745Met	c.1690G>A p.Asp564Asn	No
16429	24	M	White	33	EOSRD/LCA	Homozygous	c.2843G>Ap.Cys948Tyr		No
28566	25	F	Unknown	11	MD	Compound Heterozygous	c.498_506del p.Ile167_Gly169del	c.2843G>A p.Cys948Tyr	No
17311	26	M	Unknown	40	MD	Compound Heterozygous	c.498_506del p.Ile167_Gly169del	c.1431delG	No
19403	27	F	White	39	EOSRD/LCA	Compound Heterozygous	c.4006-1G>T N/A	c.2308G>A p.Gly770Ser	No
20736	28	F	White	16	EOSRD/LCA	Compound Heterozygous	c.2548G>A p.Gly850Ser	c.4006-10A>G	No
4441	29	F	White	46	CORD	Homozygous	c.2639A>Gp.Asn880Ser		No
22771	30	M	White	38	MD	Compound Heterozygous	c.498_506del p.Ile167_Gly169del	c.4142C>G p.Pro1381Arg	No
24139	31	M	Other	17	MD	Compound Heterozygous	c.498_506del p.Ile167_Gly169del	c.2308G>T p.Gly770Cys	No

EOSRD, early-onset severe retinal dystrophy; LCA, Leber congenital amaurosis; RP, retinitis pigmentosa; MD, macular dystrophy; CORD, cone-rod dystrophy.

**Table 2 ijms-24-13932-t002:** Quantitative OCT imaging showing an increase in foveal thickness and volume, as well as in all quadrants of the inner ring thickness (IRT) and inner ring volume (IRV), in patients with FH. There was a statistically significant difference between the two groups in the temporal inner volume.

	FH Mean (SD) *n* = 20 (65%)	No FH *n* = 11 (35%)	*p*-Value
Foveal volume	0.15 (SD ± 0.04)	0.13 (SD ± 0.03)	0.190
Foveal thickness	193.89 (SD ± 54.21)	161.13 (SD ± 36.18)	0.183
Inner ring thickness	312.26 (SD ± 38.86)	288.64 (SD ± 43.55)	0.318
Inner ring volume	0.49 (SD ± 0.06)	0.44 (SD ± 0.06)	0.104
Superior inner volume	0.51 (SD ± 0.07)	0.46 (SD ± 0.06)	0.162
Superior inner thickness	319.28 (SD ± 40.14)	292.88 (SD ± 39.62)	0.193
Nasal inner volume	0.49 (SD ± 0.07)	0.45 (SD ± 0.07)	0.109
Nasal inner thickness	312.44 (SD ± 44.56)	294.63 (SD ± 58.04)	0.395
Inferior inner volume	0.50 (SD ± 0.07)	0.45 (SD ± 0.07)	0.138
Inferior inner thickness	319.28 (SD ± 40.14)	286.88 (SD ± 43.78)	0.118
Temporal inner volume	0.47 (SD ± 0.05)	0.41 (SD ± 0.61)	0.048
Temporal inner thickness	296.22 (SD ± 32.84)	265.57 (SD ± 38.34)	0.096

**Table 3 ijms-24-13932-t003:** No significant difference was found in the rate of change (per year) in foveal thickness, foveal volume, inner ring thickness (IRT) and inner ring volume (IRV) in patients with and without FH who had more than one visit (3 months to 13 years from the baseline visit).

	Change in Foveal Thickness (μm/Year)	Foveal Volume (mm^3^/Year)	Inner ring Thickness (μm/Year)	Inner Ring Volume (mm^3^/Year)
All patients	−2.36	−0.002	0.645549	−0.002
With FH	−3.45	−0.003	−0.483	−0.0003
Without FH	−1.26	−0.001	−0.808	−0.0001

## Data Availability

The summarised data presented in this study are provided in Table 1. Full datasets are available on request from the corresponding author.

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
