# Peer review of "Foveal Hypoplasia in CRB1-Related Retinopathies"

_ijms, 2023, doi:10.3390/ijms241813932_

Round 1

Reviewer 1 Report

Rodriguez-Martinez et al. present an interesting work where they document foveal hypoplasia as part of the phenotype in a large proportion of patients (20/31, 65%) with CRB1-associated inherited retinal degenerations (CRB1-IRDs) that could undergo their structural grading. The finding was more prevalent in those with earlier symptomatic disease and supports the role that CRB1 has not only for cellular polarity and overall retinal development, but in particular, for foveal formation and foveation. Their conclusions are based on the evaluation of a large group of patients with this condition and has implications for the larger group of IRDs as well as for the prospect of treating these disorders.

My main concerns are selection biases that overestimate foveal hypoplasia in this condition, as well as the lack of a proper parametrization of their foveal hypoplasia,  which should support the findings beyond the example shown in Fig. 1. The quantitative approach, relevant to Table 2 and Figure 3, is somewhat confusing as shown – although obviously abnormal and needed. I would encourage the authors to analyze and also present data that directly relate to grading in foveal hypoplasia (Am J Ophthalmol 2012;154:779; BJO 2020;106:593), independent of the foveas being thin or not, and then relate to the sub-region volume analyses that is relevant to the general structural disorganization in this condition.

The high rate of FH is surprising high. The authors are again encouraged to use quantitative measures that unambiguously demonstrate FH independent of degeneration and to discuss possible selection biases regarding the subset of patients that could undergo their analyses. The reader will assume FH is the norm in CRB1-IRD, when it is not, at least not by inspection of the very large number of cases documented with OCT in the IRD literature.

Other suggestions:

Introduction: The authors are reminded that despite of the apparent phenotypic diversity, CRB1-associated inherited retinal degenerations remain one of the most reliably identifiable molecular forms of all inherited retinal degenerations (for example, IOVS 2011;52:6898). Recurrent features, independent of symptomatic onset, relative and variable preservation of visual acuity, or name given to the retinopathy include: 1- Foveal thinning or maculopathy, 2- Thicker than normal, often remodeled retinas in high hyperopes, 3- Relative preservation of rod-dominated peripheral retina even in severe cases. The literature insists in phenotypic diversity in CRB1-IRDs, when this recognizable pattern that likely relates to the pathophysiology of the disease, is partially shared by only a handful of other IRDs (IOVS 2018;59: 5225), the reason it maybe worth mentioning.

Introduction: Please rephrase this sentence “…representing the most to least developed fovea, whereas the atypical grade represents disruption of the photoreceptors…” Photoreceptors are not disrupted in foveal hypoplasia. Note that foveal hypoplasia has been reported in CHM (Ophthalmol. 2016;123: 2158; Ophthalmol. 2017;124:359), as an example of a disease with involvement of the RPE, no apparent connection to developmental pathways, and excellent foveal functioning.

Introduction: Not sure I would use the word ‘linked’ in this sentence “…Although FH has been linked to visual acuity…” There are examples of excellent acuities and foveal function in FH, including in asymptomatic subjects. I would also include mutations in AHR

Table 1: I assume the ‘heterozygous’ cases are compound heterozygous… Please, avoid confusion for the unfamiliar reader. Was phase confirmed? I realize this is not always possible.

Results: Figure 1 also shows RPE loss at the fovea (increased backscatter) in your example of FH, and a thinner than normal foveal in the mild example, with normal pericentral ONL, consistent with the pattern expected of CRB1-IRD. That is, foveas are nearly always abnormal, presumably either developmental abnormalities with superimposed degeneration, which explains the clinical names for the various presentations. Raises the question of whether subtle developmental abnormalities not leafing to FH may be present and triggers foveal susceptibility to degeneration (like in your mild case in Fig. 1)…

Discussion: The presence of pseudo-coloboma in ~10% of the original sample may be discussed in this setting as it may be represent the highest degree of developmental failure of the fovea. In fact, foveal hypoplasia has been reported or noted in documented images, as the mildest manifestation of the form of IRD (NMNAT1-IRD) with the with the highest prevalence of this macular abnormality (Eur J Hum Genet 2018;26:428; Retin Cases Brief Rep. 2020;16:385; Retin Cases Brief Rep. 2021;15:139).

Author Response

Reviewer: 1

My main concerns are selection biases that overestimate foveal hypoplasia in this condition, as well as the lack of a proper parametrization of their foveal hypoplasia, which should support the findings beyond the example shown in Fig. 1. The quantitative approach, relevant to Table 2 and Figure 3, is somewhat confusing as shown – although obviously abnormal and needed. I would encourage the authors to analyze and also present data that directly relate to grading in foveal hypoplasia (Am J Ophthalmol 2012;154:779; BJO 2020;106:593), independent of the foveas being thin or not, and then relate to the sub-region volume analyses that is relevant to the general structural disorganization in this condition.

DONE- Supplemental material added for more examples of FH by structural grading.

Due to a retrospective nature of the data and poor lamination seen in our cohort specific to CRB1-retinal disease, there was no consistency on measurements across different layers. Hence, the decision on classifying through qualitative measurements.

The high rate of FH is surprising high. The authors are again encouraged to use quantitative measures that unambiguously demonstrate FH independent of degeneration and to discuss possible selection biases regarding the subset of patients that could undergo their analyses. The reader will assume FH is the norm in CRB1-IRD, when it is not, at least not by inspection of the very large number of cases documented with OCT in the IRD literature.

DONE: Added to lines 313-314 selection bias

Added to lines 318-322 justification for higher rate of reported FH to limitation section in order to not mislead the reader.

Other suggestions:

Introduction: The authors are reminded that despite of the apparent phenotypic diversity, CRB1-associated inherited retinal degenerations remain one of the most reliably identifiable molecular forms of all inherited retinal degenerations (for example, IOVS 2011;52:6898). Recurrent features, independent of symptomatic onset, relative and variable preservation of visual acuity, or name given to the retinopathy include: 1- Foveal thinning or maculopathy, 2- Thicker than normal, often remodelled retinas in high hyperopes, 3- Relative preservation of rod-dominated peripheral retina even in severe cases. The literature insists in phenotypic diversity in CRB1-IRDs, when this recognizable pattern that likely relates to the pathophysiology of the disease, is partially shared by only a handful of other IRDs (IOVS 2018;59: 5225), the reason it maybe worth mentioning.

DONE- Added to lines 43-44.

Introduction: Please rephrase this sentence “…representing the most to least developed fovea, whereas the atypical grade represents disruption of the photoreceptors…” Photoreceptors are not disrupted in foveal hypoplasia. Note that foveal hypoplasia has been reported in CHM (Ophthalmol. 2016;123: 2158; Ophthalmol. 2017;124:359), as an example of a disease with involvement of the RPE, no apparent connection to developmental pathways, and excellent foveal functioning.

DONE- Sentence rephrased, paragraph 3, line 73-74

Introduction: Not sure I would use the word ‘linked’ in this sentence “…Although FH has been linked to visual acuity…” There are examples of excellent acuities and foveal function in FH, including in asymptomatic subjects. I would also include mutations in AHR

DONE- “Linked to visual acuity” removed to avoid generalisation. Line 75-76

Mutations in the AHR gene added, line 76.

Table 1: I assume the ‘heterozygous’ cases are compound heterozygous… Please, avoid confusion for the unfamiliar reader. Was phase confirmed? I realize this is not always possible.

DONE- Changed to “compound heterozygous”.

Results: Figure 1 also shows RPE loss at the fovea (increased backscatter) in your example of FH, and a thinner than normal foveal in the mild example, with normal pericentral ONL, consistent with the pattern expected of CRB1-IRD. That is, foveas are nearly always abnormal, presumably either developmental abnormalities with superimposed degeneration, which explains the clinical names for the various presentations. Raises the question of whether subtle developmental abnormalities not leafing to FH may be present and triggers foveal susceptibility to degeneration (like in your mild case in Fig. 1)

DONE- comments added on discussion paragraph 4, line 278-279.

Discussion: The presence of pseudo-coloboma in ~10% of the original sample may be discussed in this setting as it may be represent the highest degree of developmental failure of the fovea. In fact, foveal hypoplasia has been reported or noted in documented images, as the mildest manifestation of the form of IRD (NMNAT1-IRD) with the with the highest prevalence of this macular abnormality (Eur J Hum Genet 2018;26:428; Retin Cases Brief Rep. 2020;16:385; Retin Cases Brief Rep. 2021;15:139).

DONE- comments added discussion paragraph 4, line 275.

Reviewer 2 Report

This is a very interesting paper that firstly reports a type of foveal hypoplasia (FH) in eyes associated with CRB1-related retinopathy. Since the role of the CRB1 gene is the retinal development and integrity, the biggest question is how all retinal layers are preserved or altered associated with FH. I request to add some comments on this together with the following requests:

1)    Here, the grades of the associated FH varied from 1a to 1b. Please add measurements of the persistence of the inner retinal layers at the fovea in Table 1 (the FH column of Table 1 is better to show as “1a or 1b” instead of “Yes or No”).  

2)    Were there any abnormalities associated in the outer retinal layers? Please also add some comments on the appearances of the outer retina in the 4th paragraph of the Discussion.

3)    Therefore, please show as many OCT images with FH as possible that can represent the variation.

A minor comment:

Line 83: “CMO” needs a spell out (instead of line 343). Please check other abbreviations.

Author Response

Reviewer: 2

Here, the grades of the associated FH varied from 1a to 1b. Please add measurements of the persistence of the inner retinal layers at the fovea in Table 1 (the FH column of Table 1 is better to show as “1a or 1b” instead of “Yes or No”).  

DONE – A note has been added to the limitations of the study. Line 318-322

Were there any abnormalities associated in the outer retinal layers? Please also add some comments on the appearances of the outer retina in the 4th paragraph of the Discussion.

DONE- added on paragraph 4, line 271.

Therefore, please show as many OCT images with FH as possible that can represent the variation.

 DONE- Supplemental material added with more OCT examples.

A minor comment:

Line 83: “CMO” needs a spell out (instead of line 343). Please check other abbreviations.

DONE- We have spelt out the abbreviation, line 87.

Round 2

Reviewer 1 Report

The authors addressed my concerns

n/a